# A new quantitative approach to identify reworking in Eocene to Miocene pollen records from offshore Antarctica using red fluorescence and digital imaging

Stephanie L. Strother[1], Ulrich Salzmann[1], Francesca Sangiorgi[2], Peter K. Bijl[2], Jörg Pross[3], Carlota Escutia[4], Ari Salabarnada[4], Matthew J. Pound[1], Jochen Voss[5], John Woodward[1]

[1]Department of Geography, Faculty of Engineering and Environment, Northumbria University, Ellison Building, Newcastle upon Tyne, NE1 8ST, UK
[2]Marine Palynology and Paleoceanography, Department of Earth Sciences, Faculty of Geosciences, Laboratory of Palaeobotany and Palynology, Utrecht University, Heidelberglaan 2, 3584CS Utrecht, The Netherlands
[3]Paleoenvironmental Dynamics Group, Institute of Earth Sciences, Heidelberg University, Im Neuenheimer Feld 234, D-69120 Heidelberg, Germany
[4]Instituto Andaluz de Ciencias de la Tierra, CSIC-Universidad de Granada, Granada, Spain
[5]School of Mathematics, University of Leeds, Leeds LS2 9JT, UK,

*Correspondence to:* Stephanie Strother (slstrother2@gmail.com) and Ulrich Salzmann (ulrich.salzmann@northumbria.ac.uk)

**Abstract.** Antarctic palaeoclimate evolution and vegetation history after the formation of a continent-scale cryosphere at the Eocene/Oligocene boundary, 33.9 million years ago, has remained a matter of controversy. In
particular, the reconstruction of terrestrial climate and vegetation has been strongly hampered by uncertainties in unambiguously identifying non-reworked as opposed to reworked sporomorphs that have been transported into Antarctic marine sedimentary records by waxing and waning ice sheets. Whereas reworked sporomorph grains over longer non-successive geological time scales are easily identifiable within younger sporomorph assemblages (e.g., Permian sporomorphs in Pliocene sediments), distinguishing non-reworked from reworked material in palynological
assemblages over successive geological time periods (e.g., Eocene sporomorphs in Oligocene sediments) has remained problematic. This study presents a new quantitative approach to identifying non-reworked pollen assemblages in marine sediment cores from circum-Antarctic waters. We measured the fluorescence signature, including red, green and blue, brightness, intensity and saturation values, of selected pollen and spore taxa from Eocene, Oligocene and Miocene sediments from the Wilkes Land margin Site U1356 (East Antarctica) recovered
during Integrated Ocean Drilling Program (IODP) Expedition 318. Our study identified statistically significant differences in red fluorescence values of non-reworked sporomorph taxa against age. We conclude that red fluorescence is a reliable parameter to identify the presence of non-reworked pollen and spores in Antarctic marine sediment records from the circum-Antarctic realm that are influenced by glaciation and extensive reworking. Our study provides a new tool to accurately reconstruct Cenozoic terrestrial climate change on Antarctica using fossil
pollen and spores.

Keywords: Fluorescence, pollen, spores, Antarctica, reworking, vegetation, climate reconstruction

## 1 Introduction

Antarctica plays a key role in understanding past and future global climate change due to the impact its large ice
sheets exert on sea level as well as on oceanic and atmospheric circulation. Throughout the last 65 million years, the
Antarctic continent has undergone a drastic change from a greenhouse environment in the early Paleogene towards
an icehouse world in the late Paleogene and Neogene (e.g., Askin and Raine, 2000; Prebble et al., 2006; Bijl et al.,
2013; Anderson et al., 2011; Pross et al., 2012; Passchier et al., 2013). The analysis of fossil pollen and spores is one
of the most important tools for reconstructing and quantifying past vegetation and terrestrial climate change.  For
Antarctica the lack of long and well-dated sediment records puts considerable constraints on a detailed spatial and
temporal reconstruction of terrestrial environmental change. Macro- and microfossil evidence for vegetation cover
from continental sections of Antarctica is often difficult to date and in general sparse due to past and present ice
cover (e.g., Birkenmajer and Zastawniak, 1989; Pole et al., 2000; Lewis et al., 2008; Warny et al., 2016). Therefore,
most reconstructions of climate and vegetation on the Antarctic continent are based on palynological records from
marine, circum-Antarctic sediment cores. However, the waxing and waning of Antarctic ice sheets throughout the
Oligocene and Miocene caused reworking of terrestrial material into marine sediments, ultimately leading to a
combination of non-reworked and reworked palynomorphs in palaeorecords that are difficult to differentiate
especially over short geological time scales (e.g., Askin and Raine, 2000; Raine and Askin, 2001; Prebble et al.,
2006; Salzmann et al., 2011; Griener et al., 2015).

The unambiguous identification of reworked palynomorphs in palaeorecords is essential to establish reliable climate
and vegetation reconstructions for the Antarctic continent. However, a quantitative approach to differentiate non-
reworked from reworked pollen assemblages over relative short geological time scales (e.g. Oligocene to Miocene)
has not yet been established. Previous palynological studies in Antarctica have identified reworked Cenozoic
sporomorphs using the thermal alteration of grains (e.g., Askin and Raine, 2000; Raine and Askin, 2001; Prebble et
al., 2006, Griener et al., 2015; Warny et al., 2016). This approach only takes into account reworked pollen grains
that have been exposed to significantly different taphonomical conditions than the non-reworked sporomorphs.
However, submarine reworking of shelf material can only have small impacts on preservation quality, hampering the
unambiguous identification of reworked palynomorphs using light microscopy (e.g., Salzmann et al. 2011).

Subjective fluorescence microscopy has been applied in Antarctic palynology to help remedy the issue of reworking.
Raine (1998) and Salzmann et al. (2011) used autofluorescence to identify reworked Permian and Mesozoic
sporomorphs within Cenozoic sediments from the Cape Roberts cores in the Ross Sea and James Ross Island,
Antarctica. Qualitative attempts to separate reworked and non-reworked sporomorphs based on their fluorescence
colours through geological time have been shown to work (Phillips, 1972; Bujak and Davies, 1982). However, these
methods are highly subjective, being dependent upon the observer, and difficult to reproduce.

By using fluorescence microscopy this study aims to develop a new systematic and quantitative approach to identify
non-reworked pollen and spore assemblages in marine sediments from Antarctica. We measured the fluorescence
signature, including red, green, blue, brightness, intensity and saturation values, of the most common pollen and

spore taxa under ultra-violet (UV) light in Eocene, Oligocene and Miocene sediments. All samples were taken from the Wilkes Land margin sediment record at IODP Site U1356 (Fig. 1), and cover the early Eocene through the mid-

Miocene with two hiatuses from the mid- Eocene to the early Oligocene (~47 – 33.6 Ma) and from the latest Oligocene to early Miocene (~23.12 – 16.7 Ma) (Escutia et al., 2011; Tauxe et al., 2012). This provides us with four time intervals, all yielding abundant sporomorphs (Escutia et al., 2011; Pross et al., 2012, Contreras et al., 2013; Sangiorgi et al., in review) in which we hypothesise different fluorescence behaviour. The Cenozoic sediment record of Site U1356 provides a unique opportunity to compare the fluorescence of the same pollen taxa through the

Eocene to Miocene, i.e., before and during the impact of large-scale glaciation. The aim of our study is to provide a simple quantitative approach to identify the presence of non-reworked palynomorphs in a given sediment layer that can be used to reliably reconstruct past vegetation and climate for the time interval during which the sediment was deposited.

## 2 Conceptual Model for identifying non-reworked palynomorphs in Antarctic sediments

Various factors such as burial depth and geological age contribute to the fluorescence of sporomorphs. These factors reflect the ultimate determining factor of fluorescence alteration, which is heat flow and the length of time the sporomorphs are exposed to this heat (Waterhouse et al., 1998). The fluorescence signal from fossil pollen and spores comes from the sporopollenin in the exine, which contains heteroatomic compounds (Yeloff and Hunt, 2005). Over geological timescales pollen and spores in sediments are subjected to elevated temperatures and pressures after

burial, and the less resistant compounds of the sporopollenin shift to the red end of the colour spectrum and ultimately towards no fluorescence (Van Gijzel, 1967; Bujak and Davies, 1982; Yeloff and Hunt, 2005). This suggests that the colour of fluorescence changes with burial time: sporomorphs from older sediments (Cretaceous and older) show little to no fluorescence, and pollen and spores from older epochs of the Cenozoic (Paleocene, Eocene and Oligocene) show fluorescence predominantly on the red end of the spectrum with additional

fluorescence colour (orange, yellow, blue and green) variations including red fluorescence continuing through to modern (Bujak and Davies, 1982). Critically, the process of fluorescence loss is irreversible, meaning that fluorescence cannot be re-gained by the sporomorphs at any time.

The above described predictable change in colour and intensity of fluorescence in relation to burial time and depth, provides an opportunity to assess whether sporomorphs in a sediment record are non-reworked or reworked from

older strata. Antarctic Cenozoic sediments typically show a complex burial history with episodes of glacial erosion and rapid sediment deposition. The Wilkes Land Site U1356 shows shifts in the delivery of sediments to the site (e.g., shift in depocenters, changes in the amount and type of sediment delivered to the site) and erosion during two major events near the Eocene to Oligocene and Oligocene to Miocene transgression (see also Sect. 3). The alternating phases of erosion, accumulation and rapid deposition have a strong impact on the fluorescence of each

individual palynomorph and the assemblage. We addressed these factors by building our experimental approach on the following assumptions:

(i)    Each palynomorph assemblage contains a strongly varying percentage of reworked pollen and spores, which

can originate from multiple sources and ages.

(ii)   Changes in fluorescence are site- and sediment-specific, which prevents the use of fluorescence data for absolute age determinations and a comparison of the fluorescence values between sites.

(iii)  Fluorescence values in consecutive sediment layers can overlap and are strongly variable.

(iv)   Discrimination of non-reworked palynomorphs can only be based on relative measurements and not on absolute values taken from single strata and single grains.

We hypothesise that the presence of non-reworked palynomorphs in a sediment core will result in a fluorescence signal that continuously declines with decreasing depth. We further hypothesise that significantly different values between depths indicate the presence of a sufficiently high number of non-reworked pollen and spores to be used for a meaningful environmental reconstruction.

**3 Sedimentology of Site U1356 and potential source of reworking**

Pollen and spores were examined from Eocene, Oligocene and Miocene sediments from IODP Site U1356 located ~300 km off Wilkes Land, East Antarctica (63°18.6138'S, 135°59.9376'E) at the transition between the continental rise and the abyssal plain (Fig. 1; Escutia et al., 2014). The Wilkes Land margin formed during the late Cretaceous as part of a non-volcanic rift, with Oligocene-Eocene shelf sediments exposed today on the continental shelf proximal to Site U1356 (Close et al., 2009; Expedition 318 Scientists, 2011). The sediments at Site U1356 (Fig. 2) record a complex history in the development of this margin that is deeply influenced by the growth of a continental-scale ice sheet during the Eocene-Oligocene Transition (EOT) and by ice-ocean interactions since the earliest Oligocene.

During the early and middle Eocene, sedimentation took place in a shallow-water environment largely influenced by delivery of sediments from the continent. The lowermost interval (949-1006 metres below seafloor (mbsf)) dated early Eocene consists of bioturbated claystones likely deposited by hemipelagic sedimentation. Rare laminated siltstone and sandstone interbeds indicate sporadic gravity flows or bottom current activity reaching the site. Sedimentary environments during the middle Eocene (895-949 mbsf) are characterized by the presence of sandstones with clasts and contorted bedding, diamictites, and micaceous fining-upwards sandstones and siltstones, which point to deposition in a shallow-water environment. The clay fraction in all Eocene sediments contains smectite and kaolinite (Expedition 318 Scientists, 2011), which points to chemical weathering under warm and humid conditions (Expedition 318 Scientists, 2011).

During the EOT, subsidence and the glacial isostatic adjustment of the margin resulted in partial eustatic recovery on the continental shelf and erosive currents on deeper parts of the margin (Stocchi et al., 2013). At Site U1356, the non-glacial middle Eocene strata are separated from early Oligocene glacimarine strata by the WL-U3 regional unconformity (Fig. 2; Escutia et al., 2005; Escutia et al., 2011). This unconformity represents a 13 million year (Ma) hiatus (Escutia et al., 2011; Tauxe et al., 2012) suggested to predominately be caused by an extreme erosion event associated with the growth of a continental-wide ice sheet during the EOT leading to the Oi-1 event (Escutia et al.,

2011; 2014). On the continental shelf, approximately 300 - 600 m of sediment were eroded during the Oi-1 event (Eittreim et al., 1995), but partial eustatic recovery provided accommodation space for early Oligocene sediments (Escutia et al., 2011) and potentially late Eocene sediments (Stocchi et al., 2013) to accumulate.

High sedimentation rates during the Oligocene resulted in approximately 455 m (440.7 – 895.41 mbsf) of sedimentary strata being deposited at Site U1356 and dated as early to late Oligocene (~33.6 – 23 Ma) (Escutia et al., 2011; Tauxe et al., 2012). Oligocene sediments all denote deposition in deep-water environment with occasional reach of iceberg activity indicated by dropstones (Escutia et al., 2011). Sedimentation rates during the early Oligocene (723.5-895 mbsf) section are 20 m/m.y. and lithologies are characterized by interbedded laminated

claystones, bioturbated claystones and contorted diamictites and convoluted bedded mudstones (Expedition 318 Scientists, 2011). These sediments indicate that times of hemipelagic sedimentation with the influence of bottom-currents alternate with times dominated by Mass Transport Deposits (MTDs). The presence of turbidites and MTDs points to times with an increased contribution of transported sediment from shallow-marine and ultimately terrestrial sources, indicating a strong likelihood of reworked palynomorphs in the record. The late Oligocene (459.4-723.5

mbsf) is characterized by a sharp increase in sedimentation rates (89 m/m.y.). The sedimentary section is comprised by an alternation of bioturbated claystones with laminated claystones indicative of hemipelagic deposition influenced by bottom-currents of varying velocities. Interbedded with the claystones are diamictites, graded siltstones and sandstones indicative of the deposition of end members of subaqueous density flows (turbidite, debris flow, MTDs), pointing to an increase in terrigenous sediment to the site (Expedition 318 Scientists, 2011).

The late Oligocene and early to middle Miocene (23.12 – 16.7 Ma) are separated by a ~6 m.y. long hiatus (Escutia et al., 2011). The Miocene sedimentary section includes bioturbated claystones, siltstones and sandstones (Expedition 318 Scientists, 2011). The late-middle Miocene section is characterized by diatom ooze and laminated diatom-rich silty clay indicating high-biogenic and low-terrigenous hemipelagic sedimentation dominates in a relatively high productivity environment, also affected by bottom-current activity. The pebble-sized clasts and coarse sand clusters

or interbeds likely indicate ice rafting. The middle Miocene section below 278 mbsf shows an increase in bioturbation and lack of gravel-sized clasts suggesting minimal iceberg rafting. Sedimentation rates during Miocene are around 80 m/m.y. but significantly decrease after 12 Ma.

**4 Methods**

**4.1 Palynological sampling and preparation**

Pollen taxa were analysed from 28 samples between 106.62 and 998.99 mbsf (see Supplementary Table S1). All samples were processed at the Laboratory of Palaeobotany and Palynology, Utrecht University, The Netherlands, using their standard palynological processing method for marine sediments (e.g., Bijl et al., 2013). Samples were treated with 10% HCl and cold 38% HF to dissolve carbonates and silicates, respectively, and again with 10% HCl to eliminate silica gel and sieved with a 10-micrometre mesh. Residues were mounted on glass microscope

slides using glycerine jelly, and the edges were sealed with nail polish. The nail polish seems to limit the

fluorescence of the underlying palynomorphs due to the additional medium diminishing the intensity of the brightness. Therefore, we chose to consider only those palynomorphs that were away from the edges of the slide. The use of acids such as HF and HCl can alter the fluorescence of grains towards the red end of the spectra (Van Gijzel, 1971; Waterhouse, 1998). However, the same palynological processing techniques were uniformly used for all samples. 30 pollen and spore grains were randomly selected for the Miocene, late Oligocene, early Oligocene and Eocene to determine the fluorescence signatures through geological time. The aim of our study is to identify reworking in palynological assemblages over successive geological time periods. We therefore performed a pre-selection by removing obviously reworked, older-than-Eocene sporomorphs, that were extremely dark to almost opaque under a light microscope and with very little to no fluorescence under UV excitation (Fig. 3 -6a/b). We only selected those pollen and spore grains as Eocene-Miocene palynomophs that were light and translucent under a light microscope with strong fluorescence under UV excitation, and not covered by other material on the slide (i.e. organic matter). This examination was done on all slides studied from the Eocene, early Oligocene, late Oligocene and Miocene. Five common pollen and spore taxa, which are abundant in most Antarctic pollen records and also found in the majority of the Wilkes Land samples, were selected for analysis. Dependent on availability the number of different taxa per time slice varied. These taxa include (name in brackets indicates potential nearest living relative after Raine et al., 2011 and Contreras et al., 2013) *Cyathidites minor* (*Cyathea*), *Myricipites harrisii* (Casuarinaceae or Myricaceae), *Nothofagidites flemingii* (*Nothofagus*), *N. lachlaniae* (*Nothofagus*), and *Podocarpidites ellipticus* (*Podocarpus*).

**4.2 Fluorescence microscopy**

The five pollen and spore taxa were examined under light and UV-fluorescence using an Olympus BX40F microscope with a high-pressure mercury burner, dichronic mirror with a 330 – 385nm exciter filter and 420nm long-pass barrier filter. The biochemical fluorescence emitted in Cenozoic sporomorphs ranges through the red, green and blue light intensity spectrum (Bujak and Davies, 1982). Factors such as intensity, saturation and brightness also affect the fluorescence emission and were measured to test whether these variables changed with age and depth.

Pollen and spores emit fluorescence ranging from blue (400nm) to red (700nm), and the preservation of the exine helps to determine the fluorescence colour (Van Gijzel, 1971). For an initial qualitative colour classification of the investigated pollen and spores, the colour chart based on UV-fluorescence by Yeloff and Hunt (2005) was used. When correlating the UV-fluorescence signal of sporomorph grains, only comparison between the same sporomorph taxa can be done. This is due to variations in the chemical composition of the exine that affects the fluorescence colour of the grains (Hunt et al., 2007). The gain and exposures were standardised for all measured grains throughout the analysis to allow for accurate representation of the red, green and blue (RGB) values measured. For the light microscope the gain was 1.00x, and the exposure 20 ms (+2.0 EV), while under fluorescence the gain stayed at 1.70x and the exposure was 100 ms (+2.0 EV). The white balance (1.30, 1.00, 2.00) was constant through the entire process. Pictures were taken using a Nikon DS-Fil camera and analysed in image processing software

(NIS-Elements Basic Research 3.0 program). The RGB, intensity, saturation and brightness values were measured for each grain under light and UV-fluorescence. This was in relation to a greyscale from 0 (no light) to 256. The values were taken from each grain through an autodetect tool, which draws a contour around the grain. From this contour the program provides a mean value of the grain, i.e. each fluorescence pixel value from around the contour of the grain. The total fluorescence value for each individual grain was then applied to statistical evaluation.

**4.3 Statistical analyses**

To quantitatively assess the fluorescence behaviour of the five taxa from Site U1356, three different statistical approaches were used:

(i)    A Pearson's correlation coefficient (r) was calculated to determine whether the fluorescence measurements of sporomorphs correlate with age. This correlation coefficient shows the strength of the relationship from independently measured fluorescence variables against age through the Eocene to the Miocene. Coefficient values that are closer to 1 or -1 show a better linear agreement. The most statistically significant fluorescence value, red, was then measured for each taxon to indicate the correspondence of taxa red values against age. This was undertaken in IBM SPSS Statistic version 22, for determining if correlation results are statistically relevant, the highest significance level of p-values (0.01) was used.

(ii)   The Mann-Whitney *U* test was performed in PAST (Hammer et al., 2001) to compare whether two datasets that are not normally distributed are statistically different from one another. Like other marine cores from the Antarctic shelf (e.g. Askin and Raine, 2000; Anderson et al., 2011), the sediments from U1356 have a comparatively low pollen concentration, in particular in the Oligocene and Miocene sections deposited after the growth of a continental-wide ice sheet during the EOT. The overall number of palynomorphs suitable for fluorescence measurements was therefore limited and varied between samples. However, to assess statistically significant differences between samples, we have chosen the Mann-Whitney U Test, which is applicable to all samples sizes and may be used with as few as four measurements in each sample (Fowler et al. 2009). The datasets being compared are sporomorph fluorescence measurements (red, brightness and intensity) of successive geological time slices (e.g., late Oligocene vs. Miocene red values). For each value in one sample, the number of values in a second sample are counted when the value is smaller than the first sample (ties are counted as 0.5), the total of these counts are the U statistic (Hammer et al., 2001). For larger sample values (n=30), an asymptotic approximation to p-value based on normal distribution is used. Smaller values of the U statistic would support the separation of fluorescence measurements between time steps, and larger U values would support the null hypothesis indicating the groups are similar (Hammer et al., 2001). This tests if a statistically significant fluorescence signature can be identified to separate sporomorphs over subsequent geological epochs in the Wilkes Land core. The fluorescence variables red, brightness and intensity were chosen because these values had the strongest linear relationship against age (high r values) from the Pearson correlation tests.

(iii)    To determine if similar fluorescence values of palynomorphs can be grouped by age, burial depth, taxonomy or fluorescence colour, a series of 1-way Analysis of Similarities tests (ANOSIM) with 999 permutations were conducted using PRIMER 6 (Clarke and Gorley, 2006). The raw fluorescence data for the 120 measured palynomorphs was first pre-treated with a square root transformation and then a resemblance matrix was constructed using the Bray-Curtis similarity algorithm. Using ANOSIM tested if palynomorphs with similar

fluorescence values (intensity, RGB, saturation and brightness) could be grouped into categorical factors: Age – Eocene, Early Oligocene, Late Oligocene, Miocene; Burial Depth; Taxonomy (*Cyathidites minor*, *Myricipites harrisii*, *Nothofagidites flemingii*, *N. lachlaniae*, *Podocarpidites ellipticus*); fluorescence light colour (yellow, orange and red). ANOSIM tests the null hypothesis that there are no fluorescence differences between samples grouped by the levels of a factor (e.g. fluorescence values for an Eocene sample would be

distinct from a Miocene sample). If the Global R is close to 0 then fluorescence values characterised by different levels of a factor (e.g. Eocene, Early Oligocene, Late Oligocene, Miocene) are similar and the hypothesis that the age of the sample determines the fluorescence (in this example) can be rejected. Conversely, the closer the Global R value is to 1, the more strongly that factor explains the separation of the similar fluorescence values (Clarke and Gorley, 2006).

**5 Results**

**5.1 Subjective assessment of sporomorphs through fluorescence colours**

A subjective assessment of the sporomorphs fluorescence signature revealed that a purely visual estimate of fluorescence colour only allows a limited identification of reworking and separation of geological ages. Following the colour chart classification of Yeloff and Hunt (2005), each colour was assigned a number with Eocene pollen

and spores generally graded from an orange/red (46 – 49) while Oligocene grains fluoresced an orange/yellow colour (43 – 46) and Miocene grains showed similar yellow/light orange fluorescence (42 – 45) (see also supplementary material). The visual examination and identification of the fluorescence colour of a pollen or spore grain can vary dependent upon the observer, and the fluorescence colour of sporomorphs overlaps through time (Bujak and Davies, 1982). The visible red colour fluorescence clearly distinguishes Eocene sporomorphs from

Oligocene and Miocene grains in the Site U1356 material, shown from the contrast between pollen and spore grains and the slide background under red filter (Fig. 3). However, the subjective colour comparison of fluorescence alone could not distinguish between Oligocene and Miocene grains (Fig. 3; see Supplementary Table S1).

**5.2 Variation of fluorescence values through the Eocene to Miocene**

The Pearson's correlation (r) coefficient indicates a moderate to strong relationship between fluorescence values red,

intensity, brightness and geological age. Red values showed the strongest statistical correlation with age values, r = -0.46 (p < 0.0001) (Table 1a; Fig. 4a). Due to the moderate to very strong relationship between total red values and age, a Pearson's correlation was also performed on each taxon's red value (Table 1b; Fig. 4b). A very strong agreement includes *Cyathidites minor* (r = -0.69, p = 0.004) and *Nothofagidites lachlaniae* (r = -0.66, p = 0.002),

with *Podocarpidites ellipticus* (r = -0.51, p = 0.0007) showing a moderate correlation between red values and age
(Table 1b; Fig. 4b). Brightness and intensity had a moderate correlation with age, r = -0.32, but high significance (p
= 0.0003), whereas saturation values showed no relationship with age (r = -0.22). Green (r = -0.31, p = 0.0006) and
blue values (r = -0.27, p = 0.003) had a weak relationship to age with high significance. Pearson's correlation
indicates red, intensity and brightness as the most statistically significant fluorescence signatures to separate non-
reworked sporomorphs in the Wilkes Land core.

The Mann-Whitney *U* tests show statistically highly significant (p < 0.0001) changes of fluorescence red, intensity
and brightness values from the Eocene to late Oligocene and the Eocene to Miocene (Table 2). However, there is no
statistically considerable difference between the Eocene to early Oligocene red values (p = 0.2772), whereas
significant differences exist between the intensity (p = 0.0014) and brightness (p = 0.0014) (Table 2). The early
Oligocene to Miocene (p = 0.0009) and early Oligocene to late Oligocene (p = 0.0067) red signals can be
distinguished from one another (Table 2), however the intensity and brightness cannot. The late Oligocene to
Miocene red (p = 0.2772), intensity (p = 0.0451) and brightness (p = 0.0459) cannot be differentiated from each
other. This indicates that measuring the red fluorescence values from the Wilkes Land samples can distinctively
separate the fluorescence signal from sporomorphs in the Eocene, Oligocene and Miocene. The Mann-Whitney *U*
test indicates that non-successive intervals in geological time, e.g., Eocene to Miocene and Eocene to late Oligocene,
show more distinctive differences in fluorescence red, intensity and brightness values. However, unlike the
subjective fluorescence colour comparison, the Mann-Whitney *U* test shows Oligocene and Miocene grains can now
be separated based on the digital quantitative measurement of their red fluorescence signature.

**5.3 Factors potentially influencing fluorescence sporomorph signals**

To understand if geological age (e.g., Eocene, Oligocene and Miocene), fluorescence colour of palynomorphs, burial
depth and number of taxa had any influence on the similarity of fluorescence values an ANOSIM analysis was done
calculating Global R values for each factor. The ANOSIM tests demonstrated that the age (Eocene, Oligocene and
Miocene) of a sample (Global R = 0.145, P = 0.001) and the burial depth (Global R = 0.315, P = 0.001) could
explain the separation of samples with similar fluorescence into factors (see Supplementary Table S2). This shows
that both age and depth (age being a function of depth in this study) influence the fluorescence of palynomorphs.
The taxonomy of the samples (i.e. number of taxa) could not explain any similarity in fluorescence values (Global R
= 0.006, P = 0.385) and neither could the fluorescence colour of the palynomorph (Global R = 0.085 P = 0.011).

**6 Discussion**

**6.1 The advantage of applying a quantitative approach and red fluorescence**

Our approach provides a new, essential and simple tool to identify non-reworked palynological assemblages in
marine sediment records from the high latitudes that are influenced by glaciation and extensive reworking. Our
study also highlights the importance of using a quantitative approach in combination with digital imaging software
to identify non-reworked palynomorph assemblages over successive geological time scales. The quantitative

approach does not only offer a reproducible and transferable method. By detecting significant differences between the Oligocene and Miocene in the Wilkes Land core, this method has proven to be able to identify non-reworked palynomorph assemblages, which cannot be identified by using qualitative and subjective colour comparisons alone. Subjective fluorescence microscopy has been applied to separate non-reworked grains from recycled Permian-lower Mesozoic sporomorphs in the early Miocene Cape Roberts Project (CRP) 1 core (Raine, 1998). Both transmitted light (yellow to yellow-brown) and fluorescence colour (yellow to orange) comparisons could not discern Cenozoic pollen and spores (Raine, 1998; Askin and Raine, 2000; Raine and Askin, 2001). There is no apparent pattern of variation found in the fluorescence colour of sporomorphs in assemblages from the Ross Sea, emphasizing the importance of taking quantitative fluorescence measurements.

Our quantitative approach identified among the various fluorescence values (i.e. red, blue, green, brightness, saturation and intensity), the red fluorescence of pollen and spores as the most reliable indicator to differentiate reworking in successive geological epochs. From the Eocene through the Miocene, each time step shows a clear statistical grouping of red fluorescence values of pollen and spores. While individual taxa do show varying overlaps in red fluorescence values (Fig. 4b), all five pollen and spore taxa show a moderate to strong relationship with age (Fig. 4), reducing the likelihood of large amounts of reworked grains being present in the respective taxa assemblages. The Neogene sediment section of Site U1356 is expected to have significant reworking due to the submarine exposure of Eocene sediments close to the site. However, by using red fluorescence our approach was able to clearly distinguish between the Paleogene and Neogene pollen assemblages.

**6.2 Influence of heat flow, burial depth and hiatuses on fluorescence**

In order for a distinct fluorescence signature to be unambiguously identified in a palynological assemblage an ample amount of geological time between samples is needed (e.g. Van Gijzel, 1967; Bujak and Davies, 1982). It is important to discern whether fluorescence values can be distinguished over successive geological intervals and how factors such as hiatuses and burial depth can possibly affect fluorescence. The largest differences in depth and intervals of geological time in the Wilkes Land core are between the Eocene to late Oligocene and the Eocene to Miocene and these intervals show the highest significance of red, intensity and brightness values ($p < 0.0001$; Table 2). However, geological age alone does not always determine the fluorescence signal of sporomorphs. The amount and length of exposure to burial heat ultimately establishes the fluorescence alteration of sporomorphs (Waterhouse et al., 1998). Red fluorescence values are still significantly different ($p = 0.0067$; Table 2) when comparing the early and late Oligocene pollen assemblages. The oldest early Oligocene sample analysed was taken at 795.58 mbsf and the youngest late Oligocene sample was analysed at 555.19 mbsf. This difference in sedimentation rate and ultimately burial depth could contribute to the differentiation of red values between the early and late Oligocene. Burial depth is shown to play an important role in the fluorescence of sporomorphs as also indicated by the Global R value in the ANOSIM analysis (Sect. 3.3).

Disruption of sporomorph exposure to burial heat is shown to have an effect on fluorescence red values as well. Between the mid-Eocene to early Oligocene and the late Oligocene to Miocene, the differences in red values are

non-significant (p = 0.2772; Table 2). There are two major hiatuses observed in the Eocene to Miocene sediment record of Site U1356 (Escutia et al., 2011). A ~13 m.y. hiatus is found between the middle Eocene and the early Oligocene (Escutia et al., 2014). This unconformity represents extensive erosion correlating with the onset of glaciation at the Eocene-Oligocene boundary (Escutia et al., 2011; 2014; Stocchi et al., 2013). Another hiatus correlates with a regional unconformity at Wilkes Land that coincides with the Mi-1 event and extends from ~23.12 to 16.7 Ma (Escutia et al., 2005; Escutia et al., 2011). The hiatuses in the Wilkes Land core could have potentially disrupted the sporomorphs exposure to burial heat causing a less distinctive fluorescence signature between the mid-Eocene to early Oligocene and late Oligocene to Miocene. However, it is important to reiterate that our study could still find significant differences in red fluorescence between the entire Oligocene and Miocene (p = 0.0083).

**6.3 Fluorescence variation between taxa**

In order to produce comparable values for each geological time interval, our fluorescence approach requires sediment cores spanning successive geological epochs with similar palynological assemblages. The Paleogene and Neogene Antarctic palaeovegetation are unique with a number of common taxa (e.g., *Nothofagidites* and *Podocarpidites*) still present in palynological assemblages through a major climatic change from a greenhouse to icehouse world (e.g., Truswell and Macphail, 2009; Pross et al., 2012; Griener et al., 2015). When comparing the fluorescence signature between geological time intervals the same taxa must be used. Differences between the fluorescence signatures of individual taxa in the Wilkes Land assemblage are apparent (Fig. 4; Table 1). Factors such as exine composition and differential sensitivity to thermal alteration can cause variation in fluorescence measurements between sporomorph taxa (Waterhouse et al., 1998). The differences in red fluorescence signature could also have occurred due to variation in the number of taxa measured for each geological time slice. The only spore taxa in this study, *Cyathidites minor* showed the strongest correlation between fluorescence red values (r = -0.687, p < 0.004) through geological time. This could be due to the chemical composition of *Cyathidites minor*, the thickened and complex perispore (Marquez and Morbelli, 2014) or how this spore chemistry reacts to degradation over geological time.

**7 Concluding remarks: Applications and limitations of the quantitative red fluorescence approach**

The unambiguous identification of non-reworked palynomorphs is a prerequisite to fully understand the terrestrial vegetation response to periods of extensive cooling and environmental changes on Antarctica. Our study focussed on identifying non-reworked palynological assemblages from Antarctica, although this method can theoretically be applied to all palynological records. Our approach requires a sufficiently long palynological record (e.g. Miocene to Eocene) and the consistent presence of identical pollen taxa that occur in reasonable numbers in all measured time intervals. For statistical analysis, fluorescence values of samples covering successive time scales in a single core (e.g., Eocene red values against Oligocene red values) must be measured using digital imaging software. A pre-selection removing obviously reworked grains (e.g., grains older than the successive time scales that are extremely dark to opaque under a light microscope) from the analysis needs to be performed. Suitable pollen and spore taxa that can be found throughout the core over successive time scales must be determined before fluorescence analysis.

By using fluorescence microscopy this study shows that red fluorescence is the most reliable parameter to statistically identify reworking on million-year time scales during the Paleogene and Neogene. The study also

highlights intensity and brightness as sensitive indicators. It is important to emphasize that the red fluorescence values from this study are not absolute and are specific to the Wilkes Land core. Fluorescence variation between taxa is apparent.

Our study offers a new quantitative approach to identify if a sediment core contains non-reworked pollen taxa assemblages to reconstruct with high confidence Cenozoic climate change and vegetation pre- and post-Antarctic

cryosphere formation. By using the Mann-Whitney $U$ test, the approach is suitable to work with low pollen concentrations that are very common in Antarctic palynology. However, a higher number of palynomorphs and fluorescence measurement would certainly increase the potential and depth of further statistical analyses. The low pollen number, for example, prevented us from exploring whether changes in the variance of the fluorescence value can be used as a measure for the degree of reworking in each sample (i.e. high variance = high number of reworked

palynomorphs). Additional studies are needed to systematically explore the wider use of our red fluorescence approach for Antarctic palynology. These studies should include forthcoming IODP drilling expeditions and possibly existing sites such as ANDRILL (AND-2A) where several taxa such as *Nothofagidites brassii* group, Proteaceae and podocarp conifers were denoted with uncertainty because it is unknown when these taxa disappeared from Antarctica (Griener et al., 2015).

**Data availability**

The raw data generated in this study is available as supplementary material.

**The Supplement related to this article is available online at**

**Competing interests**

The authors declare that they have no conflict of interest.

**Acknowledgements**

This study was funded by the Faculty of Engineering and Environment Research Studentship from Northumbria University. The samples used for this research were provided by the Integrated Ocean Drilling Program (IODP). Ulrich Salzmann acknowledges funding received from the Natural Environment Research Council (NERC Grant NE/H000984/1). Francesca Sangiorgi thanks the Netherlands Organization for Scientific research (NWO) for NNPP

Polar 866.10.110. Peter K. Bijl acknowledges the Dutch National Organisation for Scientific Research (NWO) for VENI grant no 863.13.002. Carlota Escutia thanks the MINECO for scientific research grant CTM2014-60451-C2-1-P. Jörg Pross acknowledges support through the German Science Foundation (DFG; grant PR 651/10). We would finally like to thank the anonymous reviewer and Michael Hannah for their useful comments, which helped us to further improve this paper.

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

**Table 1. Pearson's correlation coefficient values of the (a) fluorescence variables (RGB, intensity, brightness and saturation), and (b) red values of each taxon measured, along with sample size, mean, standard deviation (SD) and p-values.**

| Fluorescent variables | N | Mean | SD | Pearson correlation coefficient (r) | p-value |
|---|---|---|---|---|---|
| Red | 120 | 29.18 | 10.08 | -0.460 | **0.0001** |
| Green | 120 | 103.51 | 35.45 | -0.308 | **0.0006** |
| Blue | 120 | 180.52 | 50.14 | -0.269 | **0.0030** |
| Intensity | 120 | 108 | 29.94 | -0.323 | **0.0003** |
| Brightness | 120 | 42.36 | 11.74 | -0.323 | **0.0003** |
| Saturation | 120 | 184.34 | 19.24 | 0.220 | 0.0158 |

| Species red values | N | Mean | SD | Pearson correlation coefficient (r) | p-value |
|---|---|---|---|---|---|
| *Podocarpus ellipticus* | 41 | 30.44 | 9.29 | -0.505 | **0.0007** |
| *Nothofagidites lachlaniae* | 19 | 28.17 | 8.03 | -0.661 | 0.0020 |
| *Nothofagidites flemingii* | 32 | 27.81 | 8.82 | -0.387 | 0.0290 |
| *Myricipites harrisii* | 13 | 41.72 | 12.03 | -0.469 | 0.1057 |
| *Cyathidites minor* | 15 | 26.29 | 8.97 | -0.690 | **0.0044** |

**Table 2. Mann-Whitney *U* test values correlating the red, intensity and brightness throughout the Eocene to the Miocene. Larger U statistic values indicate fluorescence similarity and smaller U statistic values show greater separation of fluorescence between time steps.**

| Age comparison | | Red | | | | Intensity | | | | Brightness | | | |
|---|---|---|---|---|---|---|---|---|---|---|---|---|---|
| | | U | $n_1$ | $n_2$ | p-value | U | $n_1$ | $n_2$ | p-value | U | $n_1$ | $n_2$ | p-value |
| Miocene | late Oligocene | 376 | 30 | 30 | 0.2772 | 314 | 30 | 30 | 0.0451 | 315 | 30 | 30 | 0.0459 |
| late Oligocene | early Oligocene | 266 | 30 | 30 | **0.0067** | 357 | 30 | 30 | 0.1715 | 357 | 30 | 30 | 0.1714 |
| early Oligocene | Eocene | 376 | 30 | 30 | 0.2772 | 234 | 30 | 30 | **0.0014** | 234 | 30 | 30 | **0.0014** |
| late Oligocene | Eocene | 166 | 30 | 30 | **0.0001** | 126 | 30 | 30 | **0.0001** | 126 | 30 | 30 | **0.0001** |
| Miocene | Eocene | 121 | 3 | 30 | **0.0001** | 200 | 30 | 30 | **0.0001** | 200 | 3 | 30 | **0.0001** |
| early Oligocene | Miocene | 224 | 30 | 30 | **0.0009** | 419 | 30 | 30 | 0.6520 | 419 | 30 | 30 | 0.6520 |
| Oligocene | Eocene | 293 | 3 | 30 | 0.0207 | 145 | 30 | 30 | **0.0001** | 145 | 3 | 30 | **0.0001** |
| Oligocene | Miocene | 271 | 3 | 30 | **0.0083** | 326 | 30 | 30 | 0.0679 | 326 | 3 | 30 | 0.0679 |

**Figures:**

**Figure 1.** Location of IODP Expedition 318 Site U1356 during the Eocene-Oligocene transition adapted from Houben et al. (2013). Pale blue areas indicate shelf environments and green areas show lowland regions in Antarctica. The black circle is located at 60° S. Antarctic topography and palaeoceanography modified after Lawver and Gahagan (2003) and Wilson et al. (2012).

**Figure 2.** Age-depth plot and lithostratigraphic summary for Site U1356, after Tauxe *et al.* (2012) and Escutia *et al.* (2011).

**Figure 3.** Images of Eocene, Oligocene and Miocene pollen and spore taxa analysed under white light, UV-fluorescence and red filter. The red filter shows better contrast between the grain and the background. 1. *Myricipites harrisii* (a) Miocene, Slide 15R-6W, 20-22 cm (b) Oligocene, Slide 84R-1W, 44-48 cm (c) Eocene, Slide 106R-2W, 80-83 cm, 2. *Podocarpidites ellipticus* (a) Miocene, Slide 35R-2W, 20-22 cm (b) Oligocene, 85R-3W, 20-24 cm (c) Eocene, 103R-4W, 120-124 cm, 3. *Cyathidites minor* (a) Miocene, Slide 15R-6W, 20-22 cm (b) Oligocene, Slide 50R-1W, 24-30 cm (c) Eocene, Slide 103R-4W, 120-121 cm, 4. *Nothofagidites lachlaniae* (a) Miocene, Slide 22R-2W, 20-22 cm (b) Oligocene, Slide 87R-5W, 40-44 cm (c) Eocene, Slide 97R-1W, 60-63 cm, 5. *Nothofagidites flemingii* (a) Miocene, Slide 12R-2W, 20-22 cm (b) Oligocene, Slide 59R-1W, 17-19 cm (c) Eocene, Slide 99R-2W, 40-43 cm., 6. Reworked palynomorphs not included in fluorescence measurements (a) reworked trilete spore, Oligocene, Slide 52R-CCW, 9-13cm, (b) *Classopollis chateaunovi*, Oligocene, Slide 93R-1W, 64-68 cm.

**Figure 4.** (a) All measured RGB fluorescence values for the 120 grains sampled in the Eocene, early Oligocene, late Oligocene and Miocene plotted against age. The Pearson's correlation coefficient values of fluorescence variables show the strength of the relationship between each fluorescence colour (red, green and blue) and age. (b) Red fluorescence values of each grain measured categorised by taxon (*Cyathidites minor*, *Myricipites harrisii*, *Nothofagidites flemingii*, *N. lachlaniae*, *Podocarpidites ellipticus*) plotted versus age along with Pearson's correlation coefficient to indicate the correspondence between the red values of each taxon and age.

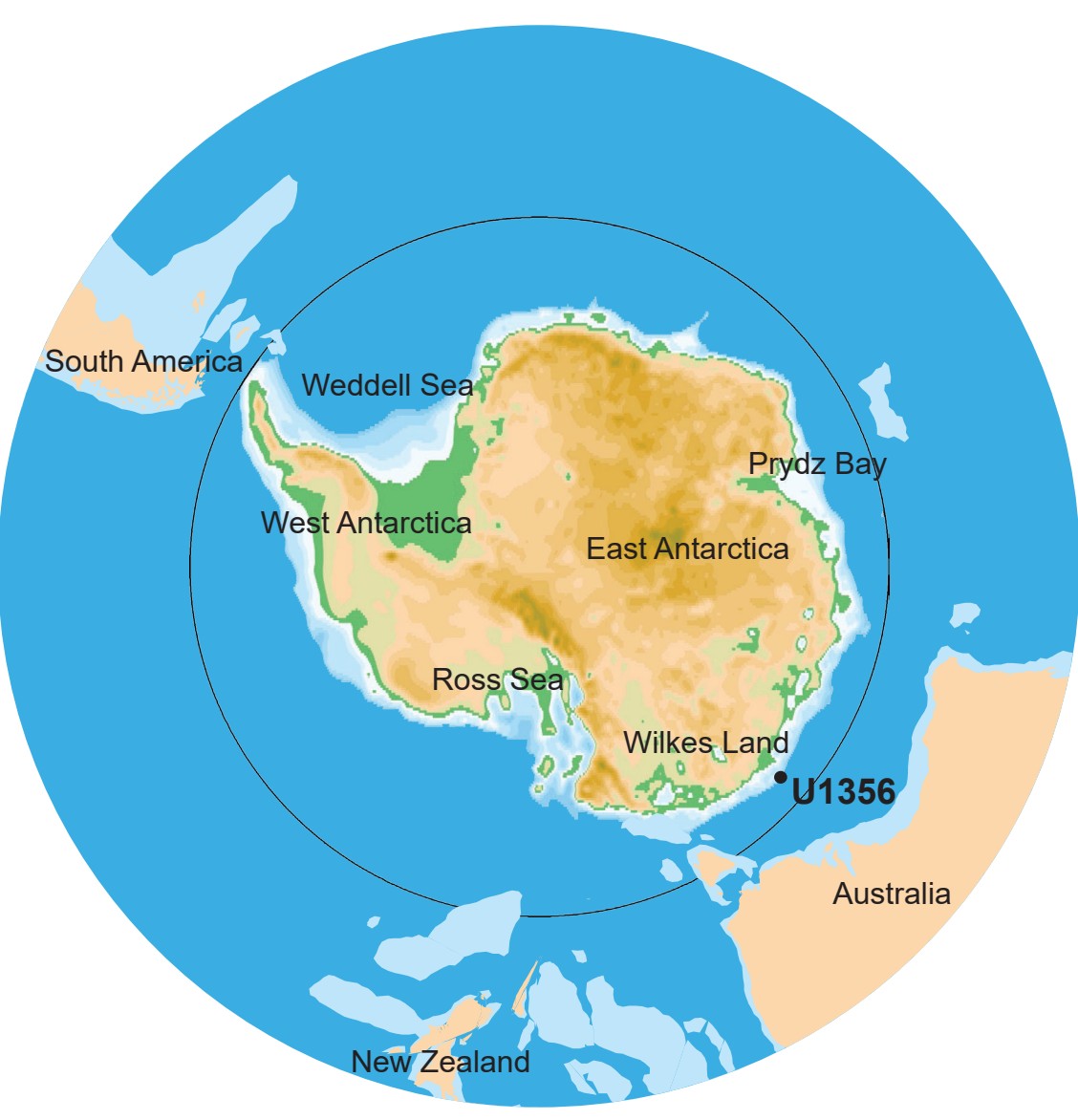

South America

Weddell Sea

Prydz Bay

West Antarctica

East Antarctica

Ross Sea

Wilkes Land

U1356

Australia

New Zealand

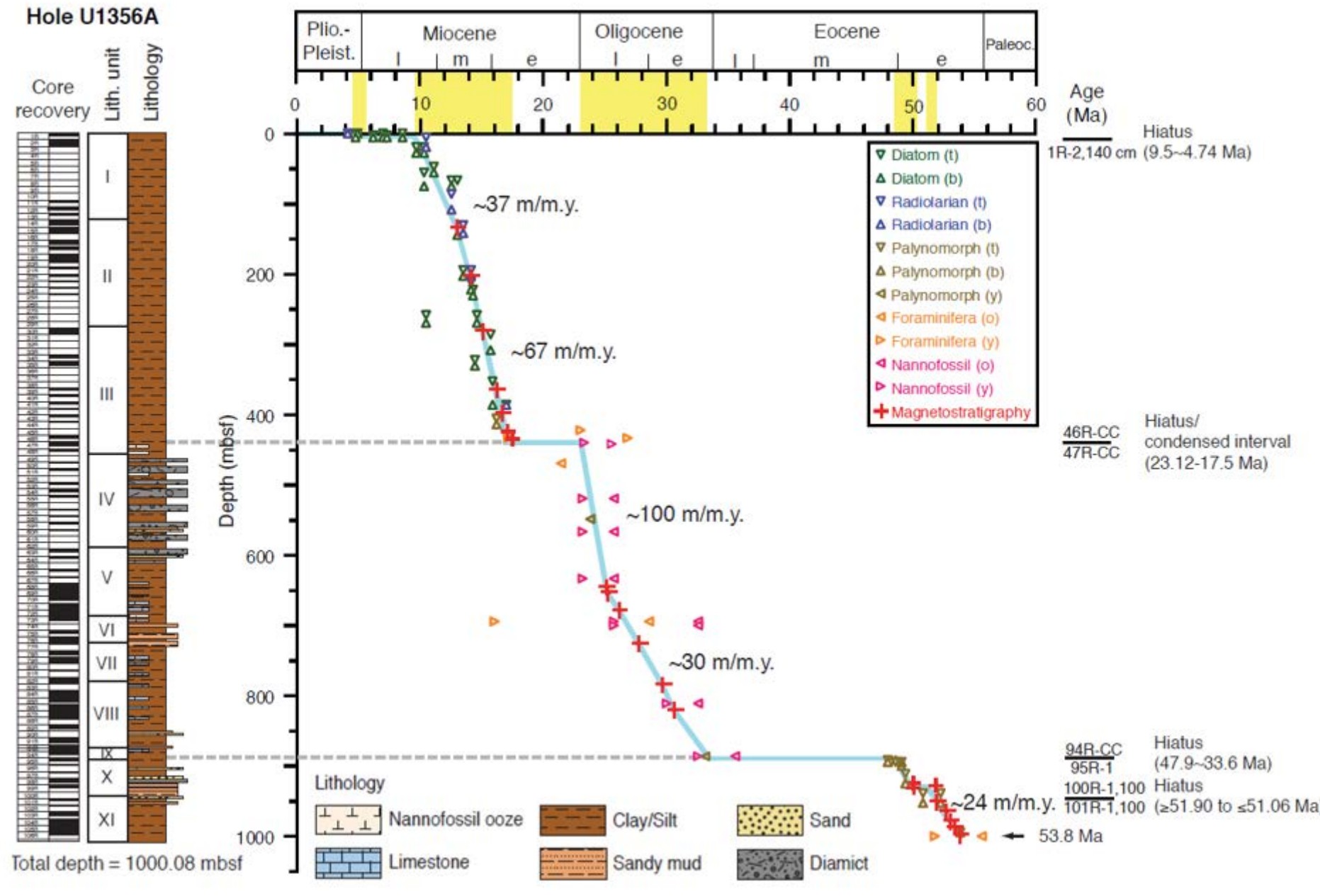

Hole U1356A

Core recovery

Lith. unit

Lithology

Total depth = 1000.08 mbsf

Plio.-Pleist. | Miocene | Oligocene | Eocene | Paleoc.

Age (Ma)

~37 m/m.y.

~67 m/m.y.

~100 m/m.y.

~30 m/m.y.

~24 m/m.y.

Diatom (t)
Diatom (b)
Radiolarian (t)
Radiolarian (b)
Palynomorph (t)
Palynomorph (b)
Palynomorph (y)
Foraminifera (o)
Foraminifera (y)
Nannofossil (o)
Nannofossil (y)
Magnetostratigraphy

Hiatus
1R-2,140 cm (9.5~4.74 Ma)

46R-CC / 47R-CC   Hiatus/condensed interval (23.12-17.5 Ma)

94R-CC / 95R-1   Hiatus (47.9~33.6 Ma)

100R-1,100 / 101R-1,100   Hiatus (≥51.90 to ≤51.06 Ma)

← 53.8 Ma

Lithology
Nannofossil ooze
Limestone
Clay/Silt
Sandy mud
Sand
Diamict

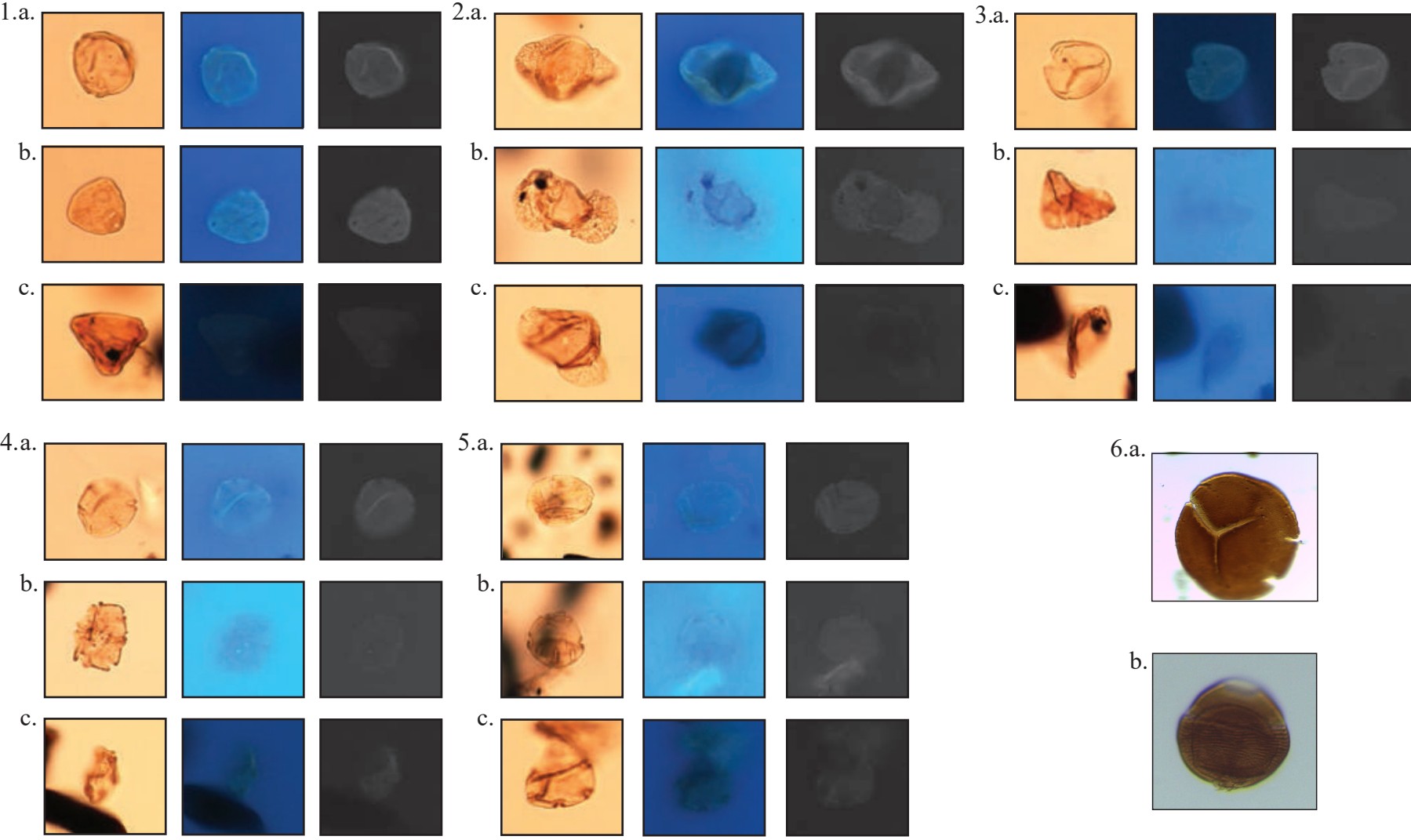

1.a.

b.

c.

2.a.

b.

c.

3.a.

b.

c.

4.a.

b.

c.

5.a.

b.

c.

6.a.

b.

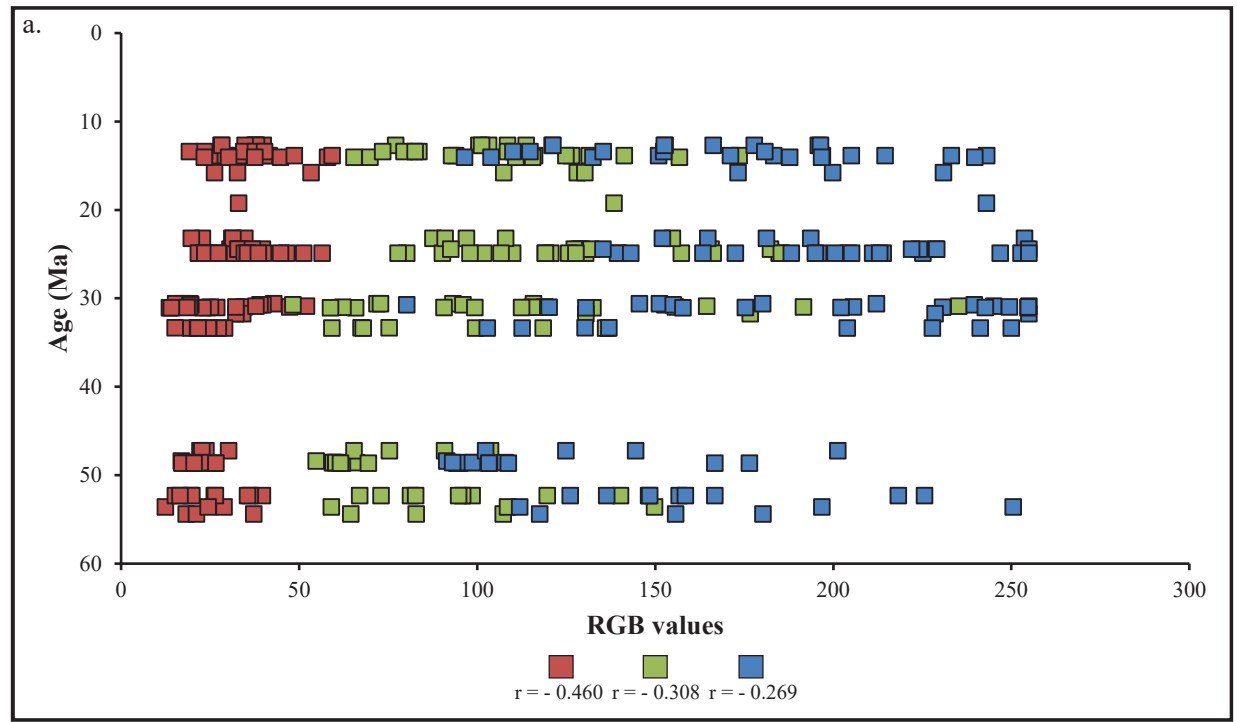

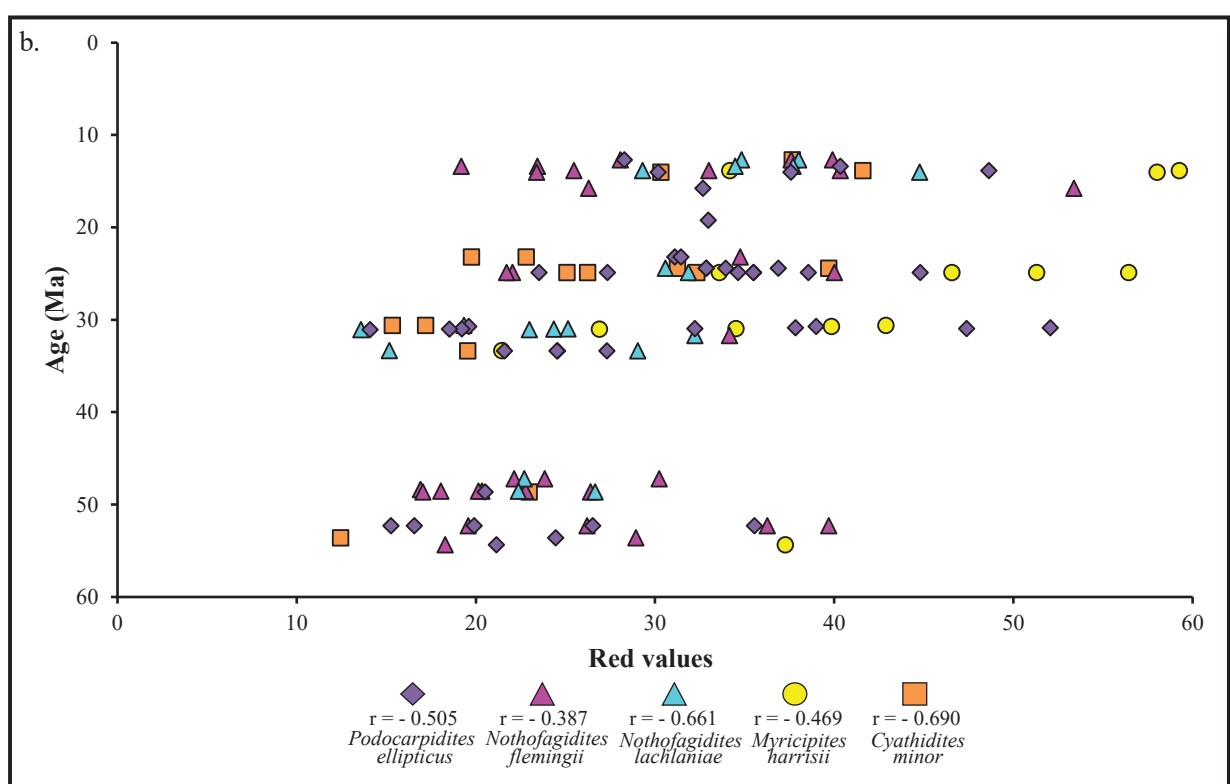