# Peer review of "A new quantitative approach to identify reworking in Eocene to Miocene pollen records from offshore Antarctica using red fluorescence and digital imaging"

_Biogeosciences, 2016_

## Referee Comment (RC1) · Anonymous Referee #1 · 9 Nov 2016

Identification of reworking in Eocene to Miocene pollen records from offshore Antarctica: a new approach using red fluorescence.

Stephanie L. Strother, Ulrich Salzmann, et al.

November 2016

This paper approaches a really important problem in post Eocene Antarctic terrestrial palynology, which is the great difficulty in distinguishing contemporaneous pollen grains from those that may have been reworked from sediments deposited during previous epochs. This difficulty results in significant uncertainty in reconstructions of post-Eocene Antarctic vegetation.

This is not a problem unique to this time or region, but is particularly pronounced in this setting due to compounding factors of 1) likely low contemporary pollen flux following large scale reduction of vegetation cover after ice sheet growth and 2) large scale reworking of sediments (and the pollen therein) due to glacial processes.

The authors use fluorescence of pollen grains as a proxy for burial history, underpinned by the observation that fluorescence intensity decreases irreversibly with a combination of temperature and time. They suggest that measurement of fluorescence parameters will allow in situ assemblages to be distinguished from those assemblages that are dominated by reworked specimens.

A really exciting advance on the use of fluorescence on Antarctic pollen, which they note has been explored before, is their extraction of RGB data and other parameters from digital images – which generates significantly richer and potentially more robust data than visual estimates of fluorescence.

I believe there are two significant problems with this paper as submitted:

1) a lack of consideration of burial history and conceptual models for reworking. If this was included, it could lead to clear and testable hypotheses to demonstrate the presence and extent of reworking (or otherwise) in this setting, and

2) sample sizes.

Burial History and conceptual models

An explicit consideration of source areas and burial history/burial depth of source material the authors have examined is lacking.

Their paper would be significantly improved by at least a conceptual model of the source of the reworked grains, and a clear hypothesis of how their results would look if there was (for example) no reworking, 50% reworking, increased reworking through time. In other words, to make the paper really useful, the reader needs to be guided more clearly on how to determine whether a new sample collected from (for example)

Site1356 contains an assemblage that is predominately reworked or contemporaneous.

At the moment, there is a significant leap of logic to get from the observation that there is a change in fluorescence parameters with depositional age, to the application of a tool to identify reworking. Demonstrating that mean fluorescence values differ between Eocene and Miocene pollen is fine, but this does not exclude the possibility that much of the Miocene assemblage is reworked.

For example, a most simple burial history possible would look something like the attached jpeg (Review Figure 1).

If Review Fig 1 was the true history, and the contemporaneous pollen flux was unchanged through time, and there was no reworking, expected fluorescence parameters down the core would decrease in a linear way (i.e. red values would decrease), due to increasing burial depth/temperature and duration down the core. The variance of the fluorescence measurements at each time step should be the same. Depending on the accumulation rate/burial depth, the mean red values from each time step could be significantly different.

But, there could be a range of effects of introducing reworked pollen grains at each time step, depending on the source and nature of these reworked grains.

The most basic reworking model is if we assume some constant proportion of grains are introduced from only the previous time step into each new sediment package (e.g. during the Miocene, 60% of the pollen deposition is contemporaneous, thus with a red value of 50, the remaining 40% is reworked from buried-then-exposed Oligocene sediments with a red value of 40). We would still expect a linear change in red values with age in the final core, and also constant variance, but variance would be larger than the first example, and the mean values slightly lower.

A more complicated example would be that a constant proportion of grains were reworked from the entire sediment column into each new sediment package (e.g. during the Miocene, 60% of pollen might be contemporaneous, thus have a red value of 50, 20% is reworked from buried-then-exposed Oligocene sediments with a red value of 40, and the final 20% is from Eocene sediments with a red value of 30). The result of this scenario would still be a slightly non-linear change in mean red value with age (slightly non-linear as some grains might get reworked more than once), and an increase in variance up the core (which incidentally is approximately what is illustrated in ms Figure 3a).

A fourth, (slightly) more realistic example would be for an increasing proportion of reworked grains into each time step, as the amount of contemporaneous input (vegetation) decreased in a step-wise fashion. The result would be (I think) an exponential increase in red values up the core, also with increased variance – which perhaps even better reflects the patterns in Ms Figure 3, particularly when the individual taxa are examined in ms Figure 3b.

These four hypothetical examples suggest that an important parameter for the identification of reworking, and the testing various hypothesised mixing/reworking models, is not identification of a strong correlation between mean fluorescence values and age, but the changes in mean and variance between time steps. It seems to me this would require much larger sample sizes than have so far been generated, but it is also not clear to me how "the presence of in situ pollen and spores" promised in the abstract can be identified without it.

A related question, one that perhaps could be answered by exploring changes in mean values between time steps as opposed to consideration of variance, is whether there is a point up the core at which all of the pollen becomes reworked (i.e. all vegetation expires). In the simple model illustrated above, assuming the removal of vegetation was abrupt and large-scale, one might expect to see a change in the slope of the line that reflects the lack of input of specimens with high red values . . . in that case, one would be looking for a "break point", rather than a regression of the entire data. As

an aside, in the analysis of comparison of mean values between time steps of the style the authors have already completed, one would imagine that one characteristic of this point could be that there was no significant difference in the means between adjacent time steps. One stated assumption, that the pollen grains that have been sampled/measured are in situ based on their preservation characteristics under light microscopy, also needs to be reframed if the authors rewrite their hypotheses more clearly.

Sample Size.

This work is under-sampled in two respects. In the first instance, 30 grains per time step is only barely sufficient to observe changes in mean values, and is certainly insufficient to explore the changes in variance from one time-step to the next required to understand the likely mixed pollen populations sampled here (Yes, 30 observations are used as a cut-off for the shift from "small" to "large" sample size in some formal statistical tests, such as the students-t distribution, but so many of the underpinning assumptions about unbiased sampling from a uniformly distributed population are just not met in the present study!).

This is compounded by the demonstration in Figure 3 that there is a clear taxonomic effect on fluorescence values – given this demonstration of such a heterogeneous pollen population (and not even considering other possibly significant causes of heterogeneity, for example facies), the previous statistical treatments in Tables 1a and 2, where all the pollen is considered as a single group, are not useful.

At the very least, the authors should ensure that the different taxa are sampled proportionally/evenly through time, but since these taxonomic effects have been identified, the authors really need to make sufficient additional observations of each taxa (or perhaps one or two target taxa) before they do further analysis.

My feeling is that a more appropriate data set for this sort of analysis would be in the order of $\sim$ 500-1500 grain measurments of a single taxa, i.e. $\sim$200-300 specimens

per time step (N. lachlaniae might be best as it is the most common, and long lived). If this is not possible (and it may well not be – there is likely not much pollen there!), more effort needs to go into describing and acknowledging the effects and implications of small sample size.

Summary

I reiterate that this is really interesting and potentially very valuable work, but it would be significantly improved by a more coherent hypothesis or conceptual model of how to apply these results, and a significantly larger data set.

Line comments

L67 suggest "are subjected to" instead of "confronted"

L70 to follow from the previous point, this needs to be qualified with something along the lines of "if burial histories are the same, fluorescence change could be used as an indicator of age"

L83 "each should come with" = "we hypothesise"?

L105 A summary of what is known of the the burial history would be helpful here – is there any constraint or estimate of the amount eroded at eh disconformities – i.e. is there any possibility the Eocene pollen was buried to a greater depth before Oligocene time etc... if these sort of effects relate to only 10s of meters of extra burial, this is useful for the reader to know

L132 modern name of Nothofagus fusca type trees has been changed to genus Fuscospora

L147 This seems sensible. So why do you then combine them for your statistics?

L157 suggest remove "in situ". . ... All you can infer is they are "not obviously reworked".. that distinction is critical for this paper!

L160 I suggest that a clear description of conceptual models of reworking is really important about here – to provide some context and reason for the statistics in the next section. . . the reasons for wanting to know why correlations against age and significant differences between mean values must be laid out.

L171 what do you mean "set" the p-value? Is this a threshold you have adopted to accept or reject a hypothesis? If so, at least this should be acknowledged/highlighted in Table 1 – perhaps bold the results with acceptable p-values?

L175 the meaning of U-values this test generates should be explained. . .. If this is a threshold score, describe what is is, where it is from and what it means, and make this clear in your Table 2 – including same comments on p-values as above

L175 Once you get into multiple sequential significance tests of this sort, perhaps describe why some sort of Bonferroni – type correction is not appropriate?

L180 Are these results tabulated?

L200 could you plot these visual data, to demonstrate there really is an advantage to using the digital data? The ranges you quote seem to overlap about as much as the fluorescence red values? The visual data does not appear in your supplementary data?

L230 The visual fluorescence data are not shown or plotted – how can you demonstrate that then that the digital measurements are better or worse at allowing differentiation of mean values between epochs?

L240 where are these results shown?

L245 following burial models discussed above – it is really not clear to me how demonstrating a linear relationship as you have done is an indicator or otherwise of reworking. This needs to be described more clearly.

L252 how? What is your threshold value or test to concluded that the sample or stage

has enough in situ pollen for reconstruction?

L259 "applied" rather than "adhered"?

[Figure]

**Fig. 1.** conceptual deposition model discussed in review

---

## Referee Comment (RC2) · M. Hannah (Referee) · 17 Nov 2016

The problem of separating reworked and in situ terrestrial palynomorphs in sediments form the Antarctic margin has hindered many palaeoenvironmental analyses of the continent. As the authors point out – while older, for example, Permian material is easily identified in younger sediments, the same is not true when the reworking is entirely confined to younger sediments. This paper offers the prospect of the development of an important tool that can separate out the reworked material.

Samples from this study were collected from three time slices in a single IODP hole from 300 km of Wilkes Land, Antarctica. The time slices are well chosen, representing important periods in evolution of the cryosphere and are the focus of many studies.

[Figure]

If the technique can be proven to work in these time intervals it will be a great step forward.

After applying standard laboratory processing techniques, thirty specimens from each sample were analysed. I am not a statistician – but this number seems low. Can the authors demonstrate that this level of counting is statistically relevant?

Five species of terrestrial palynomorphs were focused on. Were these the only ones analysed? It's not clear from the text, but it does sound like other taxa were occasionally used. Also, there has been a major revision of the genus Nothofagus, while not universally accepted, it may make a difference to the species names used if they decide to use it.

One of the concerns I have involves the determination of in situ material prior to the fluorescence analysis and its implications. As I understand the method, it is assumed that all of the specimens examined are in situ. Reworked material was identified and rejected based on a visual examination of each specimen. Of course, this is the only approach that can be used – but I think that the text should spell out clearly the limitations that this this brings to the study. I discuss this in more detail below.

As mentioned above, I am no statistician - by my entry level understanding suggests that the statistical analysis was competently done. The results are essentially summarised in figures 2 and 3. Figure 2 demonstrates the problems associated with subjective comparisons. I think that it clearly does that! Perhaps it would have been better to reduce the number of species shown and increase the size of the individual images – it may have made the point more clearly.

I have to admit, however, that I am at a loss to understand figure 3. It needs a fuller, more detailed, caption explaining exactly what was the diagram is showing – as it stands I can't make the link between the diagram and the results outlined in the text.

My uneasiness with the assumption that all the specimens measured are in situ as

outlined above is addressed to a degree in the discussion. But the argument appears to be somewhat circular. Analysis suggests that a shift to red indicates age and probable reworking, an in line 247 the authors state that the mean red fluorescence indicates that a "considerable proportion of the specimens are in situ". But how do we know that the rest aren't also in situ and the technique has failed? The authors seem to suggest that the answer lies in looking at the total assemblage and deciding whether or not sufficient numbers of individuals are in situ to trust the palynological analysis. But I'm not convinced that this gets around the circularity of the argument

The most significant issue I have with the paper is that it left me wanting more - as a practicing palynologist it appears to me that a good case has been made that measuring the mean red fluorescence data of palynomorphs offers the potential to sort out reworking of palynomorphs. But I would like to have seen a short section outlining how it may be applied in a practical sense. The inclusion of this may overcome the issues I have raised.

This work has the potential to be extremely useful, with a little expansion and clarification this paper could make an important contribution to Antarctic palynology.

---

## Author Comment (AC1) · 22 Feb 2017

We would like to thank Reviewer #1 for their very constructive comments which helped to further improve this manuscript. Before we address each "line comment" separately, we will respond to the main two concerns (1. Burial History/Conceptual Model and 2. Sample Size) Rev#1 raised in the "General Comment" section:

GENERAL COMMENTS

1. Burial History/Conceptual Model

We added two new sections to our manuscript about the approach and related hypotheses (Section 2), sedimentology (Section 3), and additional paragraphs and figure in order to address REV#1 comments regarding the missing information on burial history and the conceptual model. There are some misunderstandings which we hope to have clarified in the revised version:

a) While we agree with REF#1 that our manuscript will benefit from a more detailed description of the theoretical model and assumptions to better guide the reader on how to determine reworking, we would like to reiterate that our approach is not designed to estimate absolute ages from fluorescence values. We therefore think that conceptual models assuming linear or exponential changes, as described in Rev#1 are not useful, as they would suggest a potential to estimate absolute ages from fluorescence measurements. At least from the Oligocene onwards, the burial history of site U1356 is, like many other sites in Antarctica, dominated by repeated and partly abrupt retreats and advances of glaciers and therefore too complex to be described with a simple linear or exponential function (explained in detail in section 2 and 3).

b) Many comments made by REV#1 refer to how we compared mean values and how mean value and variance might be affected by the low sample size. There seems to be a misunderstanding as we did not compare the mean values of our fluorescence measurements. We instead avoided working with mean values as this might require a larger sample size. We might have confused REV#1 by using the term "mean" in the original manuscript for the fluorescence values produced by the imaging software. The software measures fluorescence by drawing a contour around the pollen grain and measuring multiple spots from this contour image. The program provides the mean value of the grain (i.e. each red fluorescence pixel value from around the contour of the grain). We clarified this in the revised manuscript and avoided using the term "mean". Finally, we would like to thank REV#1 for the suggestion to use variances as a tool to assess the degree of reworking. Unfortunately, our sample number is too low and variable to further explore this approach. However, we have added this suggestion to our discussion for future research.

2. Sample Size

We fully agree with Rev#1 that a much higher pollen count would be better and allow for a more detailed statistical analysis and discrimination between the

different sediment layers. Unfortunately, the Wilkes Land sediment samples have, like all other comparable Antarctic cores (e.g. ANDRILL or SHADRILL), a low pollen concentration. This is particularly the case for post Eocene samples where the pollen deposition is affected by both glacial sedimentation history and reduced pollen production on land. Measuring 500-1000 pollen grain, as suggested by Rev#1, is therefore unfortunately not possible.

However, our method is designed to work with low pollen counts and we are confident that our approach produces statistically robust results. We therefore disagree with REV#1' s comment that our work is "under-sampled" and sample number (n=30) "barely sufficient". The Mann-Whitney U Test is applicable to all samples sizes and may be used with as few as four measurements in each sample (*Fowler et al. 2009. Practical Statistics for Field Biology. 259 p.*) A methodological "disadvantage" of this test it that it reacts very sensitive to small samples sizes and normally indicates no differences if the sample size tends to be low. The fact that in our study the Mann-Whitney U Test shows statistically significant differences, despite our low sample size, gives us even higher confidence in our results and indirectly demonstrates that our sample size was actually large enough. We also think that grouping all pollen into a single group, in addition to test each single taxon, is indeed useful. With this approach we increased the sample size and, by excluding species variation, we tested one group and one explanatory variable only.

We are unsure about REV#1's comments on changes in mean value and variance as the Mann-Whitney U Test does not compare sample means. The confusion might be caused by the use of "mean values" in our manuscript, which refer to the image-processing software data (see comment above). We rephrased the relevant section to avoid further misunderstanding and also added additional paragraphs to the Method and Discussion section explaining the implications of low samples numbers for this approach as suggested by the reviewer.

LINE COMMENTS
(Line numbers are original line numbers before corrections)

*Rev#1: L67 suggest "are subjected to" instead of "confronted"*
Response: Done

*Rev#1: L70 to follow from the previous point, this needs to be qualified with something along the lines of "if burial histories are the same, fluorescence change could be used as an indicator of age"*
Response: Previous sentences have been revised to clarify the fluorescence colour can change with burial over geological timescales. However, the change of fluorescence colour cannot be used as a determination of age.

*Rev#1: L83 "each should come with" = "we hypothesise"?*

Response: Done

*Rev#1: L105 A summary of what is known of the the burial history would be helpful here – is
there any constraint or estimate of the amount eroded at eh disconformities – i.e. is
there any possibility the Eocene pollen was buried to a greater depth before Oligocene
time etc. . . if these sort of effects relate to only 10s of meters of extra burial, this is
useful for the reader to know*

Response: Following Rev#1 suggestions we added a new section, the Sedimentology (Section 3) of Wilkes Land, detailing the complex burial history, sedimentation rates and glacial influence throughout the core.

*Rev#1: L132 modern name for Nothofagus fusca type trees has been changed to Fuscospora*

Response: We are aware of the discussion started by Heenan & Smissen (2013) to split *Nothofagus* into four genera (Phytotaxa, 146 (1): 1–31). However, in order to be consistent with previous published palynological research at site U1356, Wilkes Land (e.g. Pross et al. 2012. Nature, 488, 73-77; Contreras e al. 2013. Rev Palaeobot Palyn., 197, 119-142) we prefer to keep the "old" genus name when describing the fossil record (see also discussion in Hill et al. 2015 Australian Systematic Botany, 28, 190–193).

*Rev#1: L147, This seems sensible. So why do you then combine them for your statistics?*

Response: The Pearson's correlation values were initially combined to determine which fluorescence values (red, blue, green, brightness, saturation and intensity) show a strong statistical correlation with age to further assess the fluorescence behavior of taxa. Red fluorescence showed the strongest statistical relationship with age to test against geological ages. Due to variations in the chemical composition of the exine affecting the fluorescence of grains, the red fluorescence statistical relationship was then determined for each taxon. There is a different reason to combine the red fluorescence values for the Mann-Whitney U test. As outlined under Sample Size in the General Comments section, the reason for combining the samples in the Mann-Whitney test was to increase sample size and test for one explanatory variable only.

*Rev#1: L157, suggest remove "in situ"… All you can infer is they are "not obviously reworked"… that distinction is critical for this paper!*

Response: In order to address Rev#1 concerns we replaced "in situ" in the manuscript by "non-reworked"

Reviewer: L160, I suggest that a clear description of conceptual models of reworking is really important about here – to provide some context and reason for the statistics in the next section…the reasons for wanting to know why correlations against age and significant difference between mean values must be laid out.

Response: Two additional sections detailing the burial history and conceptual model (Section 2 and 3) have been added.

*Rev#1*: L171, what do you mean "set" the p-value? Is this a threshold you have adopted to accept or reject a hypothesis? If so, at least this should be acknowledged/highlighted in Table 1 – perhaps bold the results with acceptable p-values?

Response: For determining if results are statistically relevant, we used the highest significance level of p-values (0.01), the 99 percentile. We deleted the "set threshold" and revised in Table 1 to bold results with p-values indicating the highest significant correlation (0.01).

*Rev#1*: L175, the meaning of U-values this test generates should be explained…If this is a threshold score, describe what it is, where it is from and what it means, and make this clear in your Table 2 – including same comments on p-values as above.

Response: Additional sentences have been added to section 4.3 to describe the meaning of U-values and a further explanation of U-values have been included in the Table 2 caption.

*Rev#1*: L175, Once you get into multiple sequential significance tests of this sort, perhaps describe why some sort of Bonferroni – type correction is not appropriate?

Response: The Bonferroni correction was not applied because this type of correction comes at the cost of increasing the probability of producing a false negative, i.e. reducing the statistical power of the test.

*Rev#1*: L180, Are these results tabulated?

Response: Following Rev#1 suggestion, we added a table with the ANOSIM results to the Supplementary Information (Table S2).  In addition, section 3.3 has been reworded to clarify the ANOSIM results.

*Rev#1*: L200, could you plot these visual data, to demonstrate there really is an advantage to using the digital data? The ranges you quote seem to overlap about as much as the fluorescence red values? The visual data does not appear in your supplementary data?

Response: The number associated with Yeloff and Hunt (2005) colour chart classification has been added to the supplementary material (Table S1). The ranges listed are the visual colours identified through the observations of each pollen and spore grain measured for fluorescence. Additional sentences have been added to help explain the colour classification chart and the implication of visual fluorescence colours overlapping through time.

*Rev#1*: L230, The visual fluorescence data are not shown or plotted – how can you demonstrate that then that the digital measurements are better or worse at allowing differentiation of mean values between epochs?

Response: A table with results from visual assessment has been added to the supplementary material (Table S1). Additional sentences have been added to Section 6.1 to demonstrate why digital measurements are an advantage for differentiation of mean red values between epochs. We also showed that the subjective colour comparison of fluorescence alone could not distinguish between Oligocene and Miocene grains.  We thereby demonstrated that the digital

measurement does in contrast to the visual assessment, not only produce objective and reproducible data but also more accurate results than the visual assessment. Additional sentences have been added to the discussion and result section to make this clearer.

*Rev#1*: L240, where are these results shown?

Response: ANOSIM table S2 has been added to the Supplementary Information.

*Rev#1*: L245, following burial models discussed above – it is really not clear to me how demonstrating a linear relationship as you have done is an indicator or otherwise of reworking. This needs to be described more clearly.

Response: REV#1 is correct: we used the correlation to select the best parameter. We deleted this sentence and rewrote the entire section to clarify.

*Rev#1*: L252, how? What is your threshold value or test to conclude that the sample or stage has enough in situ pollen for reconstruction?

Response: Our "threshold value" is a statistical significant difference in fluorescence colour between different pollen assemblages/depth. We added an improved and more detailed explanation of our approach in section 2 and the discussion.

*Rev#1*: L259, "applied" rather than "adhered"?

Response: Done.

---

## Author Comment (AC2) · 22 Feb 2017

We would like to thank Reviewer #2 Michael Hannah for his very constructive and helpful comments. In response to this we have taken the following actions:

*Rev#2: After applying standard laboratory processing techniques, thirty specimens from each sample were analysed. I am not a statistician – but this number seems low. Can the authors demonstrate that this level of counting is statistically relevant?*

Response: The Mann-Whitney U Test is applicable to all samples sizes and may be used with as few as four measurements in each sample (Fowler et al. 2009. Practical Statistics for Field Biology. 259 p.) A methodological "disadvantage" of this test it that it reacts very sensitive to small samples sizes and normally indicates no differences if the sample size tends to be low. The fact that in our experiment the Mann-Whitney U Test shows statistically significant differences, despite our low sample size, gives us even higher confidence in our results and indirectly demonstrates that our sample size was actually large enough.

*Rev#2: Five species of terrestrial palynomorphs were focused on. Were these the only ones analysed? Also, there has been a major revision of the genus Nothofagus.*

Response: Sentences have been updated to make clear only five common pollen and spore taxa were used for fluorescence analysis.

We are aware of the discussion started by Heenan & Smissen (2013) to split Nothofagus into four genera (Phytotaxa, 146 (1): 1–31). However, in order to be consistent with previous published palynological research at site U1356, Wilkes Land (e.g. Pross et al. 2012. Nature, 488, 73-77; Contreras e al. 2013. Rev Palaeobot Palyn., 197, 119-142) we prefer to keep the "old" genus name when describing the fossil record (see also discussion in Hill et al. 2015 Australian Systematic Botany, 28, 190–193).

*Rev#2: One of the concerns I have involves the determination of in situ material prior to fluorescence analysis and its implications. As I understand the method, it is assumed that all of the specimens examined are in situ. Reworked material was identified and rejected based on a visual examination of each specimen. Of course, this is the only approach that can be used – but I think that the text should spell out clearly the limitations that this brings to the study.*

Response: We do not assume that all of the specimen examined are "in situ". We added text and photos to Fig. 3 in order to better explain the "pre-selection process". We also added a new section 2 and additional paragraphs to the Discussion section where we provide a more detailed explanation of our approach and its limitations.

*Rev#2: I have to admit, however, that I am at a loss to understand figure 3. It needs a fuller, more detailed, caption explaining exactly what was the diagram is showing – as it stands I can't make the link between the diagram and the results outlined in the text.*

Response: The Figure 3 caption has been updated with additional explanation.

*Rev#2: My uneasiness with the assumption that all the specimens measured are in situ as outlined above is addressed to a degree in the discussion. But the argument appears to be somewhat circular. Analysis suggests that a shift to red indicates age and probable reworking,*

*an in line 247 the authors state that the mean red fluorescence indicates that a "considerable proportion of the specimens are in situ." But how do we know that the rest aren't also in situ and the technique has failed? The authors seem to suggest that the answer lies in looking at the total assemblage and deciding whether or not sufficient numbers of individuals are in situ to trust the palynological analysis. But I'm not convinced that this gets around the circularity of the argument.*

Response: The fluorescence signal of the "rest" of the specimen overlap with those from previous samples and therefore indicate that the sporomorphs have been reworked into the younger layers. A considerable portion of non-reworked sporomorphs is required to produce a signal which is significantly different from the previous. The principle of our approach is not fundamentally different from previous methods using autofluorescence in Palynology. We just propose using digital imaging (to measure the signal) and statistical analysis to make this process reproducible and independent from subjective classification of the analyst. We hope to have clarified our approach by adding a new section (Sect 2) to better explain the conceptual model and underlying assumptions.

*Rev#2: But I would like to have seen a short section outlining how this approach may be applied in a practical sense.*

Response: The practical applications have been addressed in the newly added concluding remarks (7) and section 2, where we explain the conceptual model and underlying assumptions.

---

## Author Response (AR2)

**We addressed all comments listed in the Editor's letter:**

*Line 100: These does not have to be bold as part of the paragraph.*

Done

*Chapter 5.2: some number have been changed to only two decimal places, other not, I don't see the logic. Perhaps 2 is enough?*

Done. R changed to 2 decimal places.

*Chapter 6: make sure to remove 4 (also at 6.1; 6.2 etc)*

Checked

*Chapter 6: Check the title. .... applying a quantitative approach such as red fluorescence.*

Title revised

*Line 369: a , between new and essential*

Done